# TOWARD LEARNING GEOMETRIC EIGEN-LENGTHS CRUCIAL FOR ROBOTIC FITTING TASKS

## ABSTRACT

Some extremely low-dimensional yet crucial geometric eigen-lengths often determine whether an object can be fitted in the environment or not. For example, the *height* of an object is important to measure to check if it can fit between the shelves of a cabinet, while the *width* of a couch is crucial when trying to move it through a doorway. Humans have materialized such crucial geometric eigen-lengths in common sense since they are very useful in serving as succinct yet effective, highly interpretable, and universal object representations. However, it remains obscure and underexplored if learning systems can be equipped with similar capabilities of automatically discovering such key geometric quantities in doing robotic fitting tasks. In this work, we therefore for the first time formulate and propose a novel learning problem on this question and set up a benchmark suite including the tasks, the data, and the evaluation metrics for studying the problem. We explore potential solutions and demonstrate the feasibility of learning such eigen-lengths from simply observing successful and failed fitting trials. We also attempt geometric grounding for more accurate eigen-length measurement and study the reusability of the learned geometric eigen-lengths across multiple tasks. Our work marks the first exploratory step toward learning crucial geometric eigen-lengths and we hope it can inspire future research in tackling this important yet underexplored problem.

## 1 INTRODUCTION

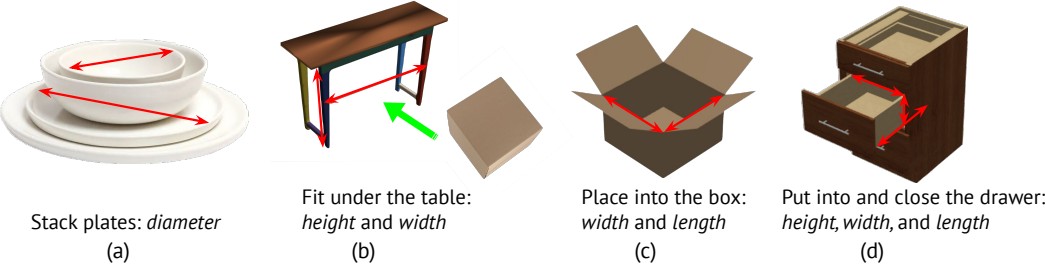

Stack plates: *diameter* (a)  Fit under the table: *height* and *width* (b)  Place into the box: *width* and *length* (c)  Put into and close the drawer: *height*, *width*, and *length* (d)

Figure 1: Example tasks and the hypothesized crucial geometric measurements by humans.

Consider a robot tasked with placing many small objects on warehouse shelves, where both the objects and the shelves have diverse geometric configurations. While the robot can simply try to accomplish the task by trial-and-error, to us as humans, it is clear that certain placements should not be attempted because they will obviously fail. For example, we should not attempt to place a tall object on a shelf whose height is too low. We base this judgement on the estimation of a critical geometric eigen-length or measurement, the *height* of the object and the shelf, whose comparison allows a quick estimate of task feasibility.

While object *height* is an example of important eigen-lengths of an object that is crucial for the above shelf placement task, it is not hard to think of many other types of object eigen-lengths for other fitting tasks. Figure 1 presents some other example tasks together with the presumable geometric eigen-lengths based on human common sense. For example, the geometric eigen-length *diameter* is important for the task of stacking plates in different sizes (Figure 1, (a)), while the *width* and *length*

of an object are crucial geometric eigen-lengths for deciding if one can put an arbitrary shape object into an open box (Figure 1, (c)).

Having such extremely low-dimensional yet crucial geometric eigen-lengths extracted as the representations for objects is certainly beneficial for designing learning systems for robotic fitting tasks. One telling evidence is that we humans have naturally built up the vocabulary of geometric key quantities, such as *height, width,* and *diameter*, when perceiving and modeling everyday objects, and used them to perform various object fitting tasks. Besides being succinct yet effective abstractions of objects for quickly estimating the feasibility for the downstream fitting tasks, such crucial geometric eigen-lengths are also highly *interpretable*, which exposes the principled reasoning process behind the feasibility checking, and *universal*, as they are generally applicable to objects with arbitrary shape and useful across different downstream tasks.

Current research in representation learning for computer vision and robotics has mostly been focusing on learning high-dimensional latent codes or injecting human knowledge as inductive bias for learning structured representations. While learning high-dimensional latent codes provides total flexibility learning any useful feature for mastering the downstream tasks, these latent codes are high-dimensional, hard to interpret, and may be prone to overfitting to the training domain. For structured representations, though researchers have explored using different kinds of object representations, such as bounding boxes (Tulsiani et al., 2017) and key points (Manuelli et al., 2019), to accomplish various downstream tasks in computer vision and robotics, these structure priors are manually specified based on human knowledge about the tasks. In contrast, we aim to explore the automatic discovery of low-dimensional yet crucial geometric quantities for robotic fitting tasks while injecting the minimal human prior knowledge – only assuming that we are measuring eigen-lengths of the input objects.

In this paper, we first propose to study a novel learning problem on discovering low-dimensional geometric eigen-lengths crucial for fitting tasks and set up the benchmark suite for studying the problem. As illustrated in Figure 2, given a fitting task (putting the bowl inside the drawer of the table) that involves a scene geometry (the table) and an object shape (the bowl), we are interested in predicting whether the object can fit in the scene accomplishing the task or not, via discovering a few crucial geometric eigen-lengths and composing them into a task program which outputs the final task feasibility estimation. To study the problem, we also define a set of commonly seen robotic fitting tasks, generate large-scale data for the training and evaluating on each task, and set up a set of quantitative metrics for evaluating and analyzing the method performance and if the emergent geometric eigen-lengths match the desired ones humans usually use.

We also explore potential solutions to the proposed learning problem and present several of our key findings. First of all, we will show that learning such low-dimensional key geometric eigen-lengths are achievable from only using weak supervision signals such as the success or failure of training fitting trials. Secondly, the learned crucial geometric eigen-lengths can be more accurately measured if geometric grounding is allowed and attainable for certain fitting tasks. Finally, we make an initial stab at exploring how to share and re-use the learned geometric eigen-lengths across different tasks and even for novel tasks. Marking the first step defining and exploring this important yet underexplored problem, we hope our work can draw people's attention to this task and inspire future research in designing solutions tackling it.

To summarize, this work makes the following contributions:

- We propose a novel learning problem on discovering low-dimensional geometric eigen-lengths crucial for fitting tasks;
- We set up a benchmark suite for studying the problem, including a set of fitting tasks, the dataset for each task, and a range of quantitative and qualitative metrics for thorough performance evaluation and analysis;
- We explore potential solutions to the proposed learning problem and present some key take-away messages summarizing both the successes and unresolved challenges.

## 2 RELATED WORK

**Learning Geometry Abstraction.** A long line of research has focused on learning low-dimensional and compact abstraction for input geometry. Given as input a 2D or 3D shape, past works have studied learning various geometric abstraction as the shape representation, such as bounding boxes (Tulsiani et al., 2017; Sun et al., 2019), convex shapes (Deng et al., 2020), Gaussian mixtures (Genova et al.,

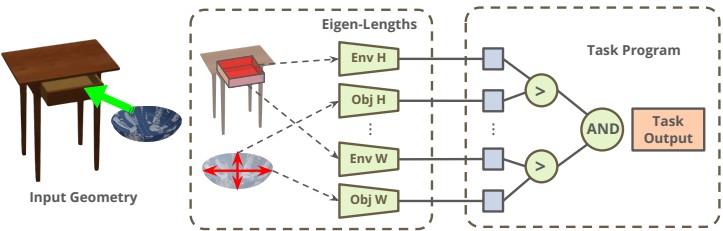

Figure 2: **Proposed Learning Paradigm** where we first predict a set of geometric eigen-lengths from the input geometries, then compose them using a task program to get the final task output.

2019; 2020), superquadrics (Paschalidou et al., 2019; 2020), parametric curves (Reddy et al., 2021) and surfaces (Sharma et al., 2020; Smirnov et al., 2020), *etc.*. Most of these works use geometry fitting as the primary objective. Our work, however, focuses on discovering geometric abstraction that can help solve the downstream manipulation tasks instead of reconstruction.

There are also previous works exploring ways to learn task-specific geometry representation for manipulation tasks. For example, researchers have tried to learn key points (Manuelli et al., 2019; Qin et al., 2020; Wang et al., 2020; Chen et al., 2020; Jakab et al., 2021; Chen et al., 2021) and affordance information (Kim & Sukhatme, 2014; Mo et al., 2021a;b; Turpin et al., 2021; Deng et al., 2021) for robotic manipulation tasks. These works mostly pre-define the types of geometry abstraction and the downstream policies to use the extracted shape summaries, and the abstraction is mostly dense or high dimensional. In this paper, we aim for useful geometric eigen-lengths and ways to automatically discover and compose them for solving manipulation tasks.

**Disentangled Visual Representation Learning.** Another line of work focuses on unsupervised representation learning techniques that pursue disentangled and compositional latent representations for visual concepts. For example, InfoGAN (Chen et al., 2016), beta-VAE (Higgins et al., 2017), and many more works (Higgins et al., 2016; Siddharth et al., 2017; Yang et al., 2020) discover disentangled features, each of which controls a certain aspect of visual attributes, usually with reconstruction as the objective. In contrast to their primary objectives of controllable reconstruction or generation, we explore the problem of learning geometric eigen-lengths driven by the goal of accomplishing downstream fitting tasks. Also, our task involves reasoning over two geometric inputs and comparing the extracted eigen-lengths on both inputs, while these previous works on disentangled visual representation learning factor out visual attributes for a single input datum.

## 3 LEARNING PROBLEM FORMULATION

Given a robotic fitting task $T \in \mathcal{T}$, we aim to learn very few but the crucial geometric eigen-lengths $\mathcal{L}_T$ (*e.g.*, *width, length, height*) of the object shape $O \in \mathcal{O}$ and the environment geometry $E \in \mathcal{E}$ that are useful for checking the feasibility of fitting $O$ into $E$ under the task $T$. Figure 2 presents an example of the proposed learning problem where the task is to put the bowl ($O$) inside the drawer of the cabinet ($E$). In this example, the *width, length, height* of the drawer and the bowl are the crucial desired geometric eigen-lengths ($\mathcal{L}_T$) and we can compose them in a task program to output the final task feasibility prediction. We consider each eigen-length $L \in \mathcal{L}_T$ as a function mapping from the input object shape $O$ or the environment geometry $E$ to a scalar value for the eigen-length measurement, *i.e.* $L : \mathcal{O} \cup \mathcal{E} \rightarrow \mathbb{R}$. After obtaining the eigen-length measurements for both the object and environment inputs, *i.e.* $\{L(O) | L \in \mathcal{L}_T\}$ and $\{L(E) | L \in \mathcal{L}_T\}$, we perform pairwise comparisons between the corresponding eigen-lengths checking if $L(O) < L(E)$ holds for every $L \in \mathcal{L}_T$. The task of fitting $O$ in $E$ under the task $T$ is predicted as successful if all the conditions hold and as failed if any condition does not hold. This format of task program is based on the intuition that in fitting tasks, we require the object to be "smaller" than the parts of the environment affording the action. Durining training, the learning systems see many fitting trials over different objects and environment geometric configurations together with their ground-truth fitting feasibility, *i.e.* $\{(O_i, E_i, \text{Successful/Failed}) | i = 0, 1, 2, \cdots \}$. The goal is to learn eigen-length functions based on which correct prediction of task feasibility given test input $(O_{test}, E_{test})$ can be made.

# 4 CAN GEOMETRIC EIGEN-LENGTHS BE LEARNED FROM BINARY TASK SUPERVISION?

In this work, we are interested in learning geometric eigen-lengths that are crucial for downstream tasks. We hope to achieve automatic discovery of these eigen-lengths from doing tasks as it requires the least human prior and allows maximum flexibility. Therefore, we start with the minimum form of supervision and explore the following question: given only binary task success/failure supervision, is it possible to learn geometric eigen-lengths of input geometries that are sufficient for the task?

## 4.1 TESTBED FOR EIGEN-LENGTH LEARNING

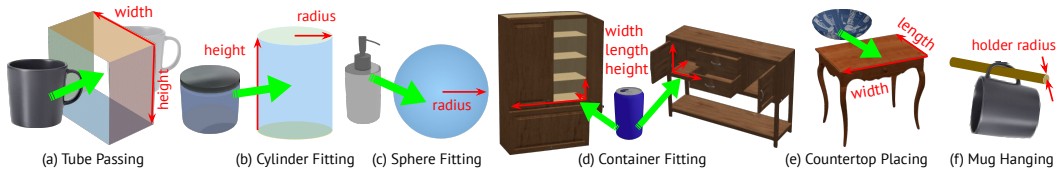

Figure 3: **Summary of tasks** and their human-hypothesized key measurements/eigen-lengths.

We start by curating a set of tasks as the testbed for the learning problem, as summarized in Fig. 3. For each task, we build a large-scale dataset comprising diverse shapes and configurations.

**Task Design Principles**  We design the tasks to (1) cover a wide range of geometries, including synthetic, simple primitive shapes and more complex ones like ShapeNet objects; (2) facilitate the analysis and interpretation of learned eigen-lengths. Specifically, here we base the analysis on comparisons to human-hypothesized eigen-lengths: given a task, humans can identify related key eigen-lengths (referred to as "ground truth" in the following), *e.g.*, object height when putting them on shelves. Comparing the learned eigen-lengths to these "ground truth" may provide important insights. To achieve this, we need accessible ground truth eigen-lengths to begin with. Primitive shapes like cylinders are ideal as they are parameterized by eigen-lengths like radius and height.

**Task Specifications**  In all tasks, we aim to determine whether a placement/motion of the object exists in a certain environment, specifically:

(a) **Tube passing. (Tube)** Pass an object through a rectangular tube. A *tube* is a cuboid without the front and back faces. Width and height of the tube/object are the key eigen-lengths to compare.

(b) **Cylinder fitting. (Cylinder)** Place an object into a cylindrical container. Bounding sphere radius of the object in XY plane and its height, as well as the radius and height of the cylinder container are the key eigen-lengths.

(c) **Sphere fitting. (Sphere)** Place an object into a spherical container. Radius of the bounding sphere of the object and the container is the key eigen-length.

(d) **Container fitting. (Fit)** Place an object into cavities in another ShapeNet container object. Example *cavities* include drawers or shelves (See Fig. 3d) of furniture. Most cavities have cuboid-like shapes. Thus, key eigen-lengths are width, length and height of cavities and objects.

(e) **Countertop placing. (Top)** Place an object on top of another ShapeNet environment object, such that its projection along the gravity axis is fully enclosed by the environment countertop. Width and length of the countertop surface and the object are key eigen-lengths.

(f) **Mug hanging. (Mug)** Hang a mug on a mug holder by its handle. The holder is a cylinder-shaped rod. Key eigen-lengths are the distance between sides of the mug handle and the diameter (or equivalently, the radius) of the mug holder rod.

**Data Generation Details**  For objects to be fitted in tasks **(a)-(e)**, we use ∼1200 common household object models from 8 training and 4 testing categories in ShapeNet (Chang et al., 2015), following Mo et al. (2021b). During data generation, we apply random scaling to the object model, then sample $N = 1024$ points from the object surface. Note that we also apply a random rotation to the object. In **(d)**,**(e)**, we use furniture and appliances from ShapeNet as the environment geometry, including ∼550 shapes from 7 object categories. In **(f)**, we use ∼200 ShapeNet mugs. We randomly sample the parameters of primitive shapes and the scaling factors of ShapeNet shapes, then sample $M = 1024$ points from their surfaces. For all tasks, we generated 75k training and 20k testing environment-object pairs. Please refer to Appendix A.3 for more data generation details.

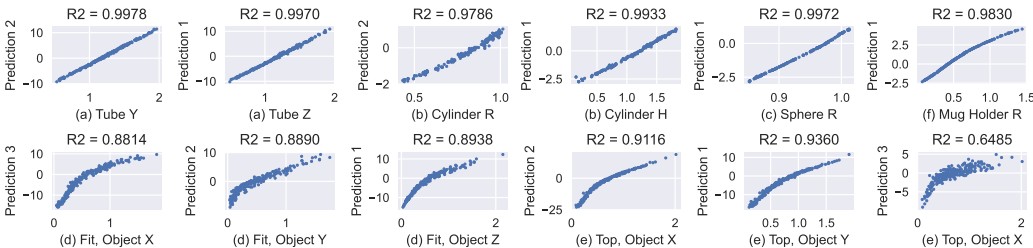

**(a)**                                                    **(b)**

Figure 4: **Network architectures.** (a) A minimal eigen-length learning pipeline where we separately encode environment and object into eigen-length values, perform pair-wise comparison, and take the logical AND of results. (b) A geometry-grounded framework where we first predict vectors and points as the geometry grounding, then compute eigen-lengths from them.

Figure 5: **Correlation Analysis.** Each plot shows the relationship between one learned eigen-length (Y coord.) and its matching "ground truth" measurement (X coord.). Higher $R^2$ values imply a stronger correlation.

## 4.2  A MINIMAL NETWORK ARCHITECTURE

Intuitively, we can measure the object and the environment separately and see if the object is "smaller" than the environment. Thus we come up with the minimal network architecture shown in Fig. 4 (a). We separately map the object and environment geometries into two sets of eigen-lengths, perform pairwise comparisons between them, and compose comparison results using logical AND.

Concretely, we encode object point cloud $O$ and environment point cloud $E$ using two Point-Net (Qi et al., 2017) networks, ObjNet and EnvNet. Both networks output $S$-dim vectors $\vec{L^{obj}} = (L_1^{obj}, L_2^{obj}, \ldots, L_S^{obj}), \vec{L^{env}} = (L_1^{env}, L_2^{env}, \ldots, L_S^{env})$. We then compute task success as $\hat{T}(E, O) = \bigwedge_{s=1}^{S}[L_s^{env}(E) > L_s^{obj}(O)]$. During training, we use a differentiable approximation $\tilde{T}(E, O) = \prod_{s=1}^{S} \sigma((L_s^{env}(E) - L_s^{obj}(O))/\tau)$, where $\tau$ is a learnable parameter. We set $S = 1$ for **(c)** Sphere, **(f)** Mug, $S = 2$ for **(a)** Tube, **(b)** Cylinder, $S = 3$ for **(d)** Fit, **(e)** Top.

## 4.3  ANALYSIS OF LEARNED EIGEN-LENGTHS

We analyze the eigen-lengths learned by the network by comparing them to "ground truth" eigen-lengths as shown in Fig. 3. For each task, we randomly sample $N = 512$ test data points and obtain the corresponding $N$ eigen-length predictions $\{L_{s,i}^{pred}\}_{i=0,\ldots,N-1}$ for each of the $S$ learned eigen-lengths, as well as $N$ values $\{L_{s',i}^{gt}\}_{i=0,\ldots,N-1}$ for each of the $S'$ "ground truth" eigen-lengths.

For each pair of predicted and "ground truth" eigen-lengths $(s, s')$, we draw a scatter plot of points $(L_{s',i}^{gt}, L_{s',i}^{pred})$ and perform least squares linear regression over them to get corresponding $R^2$-scores. We match the predictions and groundtruths by maximizing the sum of $R^2$-scores and show the scatter plots of matched pairs in Fig. 5. Note that in **(e)** Top, since we predict $S = 3$ eigen-lengths while there are only $S' = 2$ groundtruth eigen-lengths, we show the unmatched prediction with its most correlated groundtruth. For complete $S \times S'$ plots, please refer to Appendix B.2.

**Learned eigen-lengths are strongly correlated with human-hypothesized measurements.** As Fig. 5 shows, $R^2$ values between predictions and "ground truths" are close to or greater than 0.9 except for the redundant prediction slot 3 in **(e)** Top. They also have clear one-to-one correspondences with ground truth in tasks with multiple eigen-lengths, suggesting good disentanglement is learned.

**Knowing the number of eigen-lengths beforehand is not a requirement for successful learning.** The number $S$ of eigen-lengths to learn is a hyperparameter set before learning. However, it does

not have to be the exact number of relevant eigen-lengths. As shown in **(e)** Top, when we have more slots for eigen-lengths than needed, "ground truth" eigen-lengths are still captured by the first two predictions. The third prediction does not strongly correlate with any "ground truth". A further probe reveals that comparisons of this eigen-length almost never (only in $0.4\%$ of the cases) contribute to the final result, outputting `True` most of the time. The network learns a pair of degenerate eigen-lengths as there is no more necessary information to capture.

## 5    CAN GEOMETRY GROUNDINGS BE DISCOVERED FOR EIGEN-LENGTHS?

While Fig. 5 shows strong correlation between learned eigen-lengths and "ground truth", their relationship is not always perfectly linear, as can be observed in **(d)** Fit and **(e)** Top with complex geometries. Even in more linear cases, the scaling and offset make the raw eigen-length value hard to understand, *e.g.*, negative "length" values are less intuitive. As eigen-lengths can be seen as measurements of the object, many of them have sparse supports or geometry groundings on the objects, *e.g.*, height is the distance between the base plane supporting the object and its highest point. These geometry groundings anchor the corresponding eigen-length values, provide an intuitive explanation of these values, and usually carry geometric/semantic importance themselves. We are therefore interested in the following question: can we ground the eigen-lengths on geometry? From a high level, instead of directly predicting eigen-length values, if we first predict some geometric entities like points, vectors, and planes, then derive eigen-lengths from them, is it possible to learn meaningful eigen-lengths and geometry groundings?

### 5.1    GROUNDING EIGEN-LENGTH PREDICTIONS ON GEOMETRIC PRIMITIVES

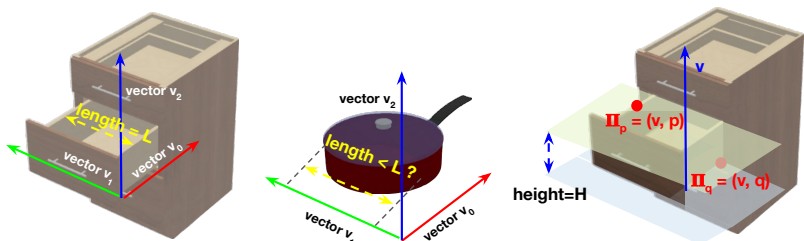

Figure 6: **Eigen-Length Geometry Groundings.** We ground each eigen-length $L$ with a unit vector $v$ and two parallel planes $\Pi_p, \Pi_q$ with normal $v$. $L$ is computed as the distance between $\Pi_p, \Pi_q$.

Consider fitting tasks like **(d)** Container Fitting and **(e)** Countertop Placing where the spaces affording the task can be roughly described by a set of parallel planes. [1] To compute the success label of the task, say fitting an object into a nightstand, we can measure the size of the spaces of interest in the environment (the drawer part) along important directions (its main axes) and comparing it to the measurement of the object. Inspired by this, we ground a pair of eigen-lengths on a tuble of unit vector and two planes $(\vec{v}, \Pi_p, \Pi_q)$ as illustrated in Fig. 6: we measure both the object and the environment along $\vec{v}$. We take the object measurement as the diameter of the projection of the object point cloud $O$ on the vector $\vec{v}$, *i.e.* $L^{obj}(O) = \max_{p \in O} \vec{v}^T p - \min_{p \in O} \vec{v}^T p$. For the environment, we use a pair of parallel planes $\Pi_p, \Pi_q$ with normal $\vec{v}$ to separate out a certain region relevant to the task (the drawer), then measure the distance between the planes. In practice, we adopt the (point, normal) plane representation and predict a point pair $(p, q)$ that determines the plane pair. The environment eigen-length is then computed as $L^{env}(E) = \vec{v}^T(q - p)$.

Figure 4 (b) illustrates our network architecture. In **VectorNet**, we employ a PointNet classification backbone to extract global feature of the environment point cloud $E \in \mathbb{R}^{M \times 3}$, then use an MLP to predict $S$ 3D vectors $\{\vec{v}_s\}_{s=1,2,...,S}$. In **WeightNet**, we employ a PointNet segmentation backbone to extract per-point features, then use $S \times 2$ MLPs with to predict $S$ pairs of probability distributions $W_s^p, W_s^q$ over the point cloud. The point coordinates of $(p_s, q_s)$ are then computed as the weighted average of original point cloud coordinates, namely $p_s = W_s^{pT}E, q_s = W_s^{qT}E$.

---

[1]Note that other tasks may require other inductive bias. We focus on this type of tasks to study the feasibility of geometry-grounded eigen-length learning. We leave a more versatile system as future work.

## 5.2 ANALYSIS OF LEARNED GEOMETRIC PRIMITIVES AND EIGEN-LENGTH VALUES

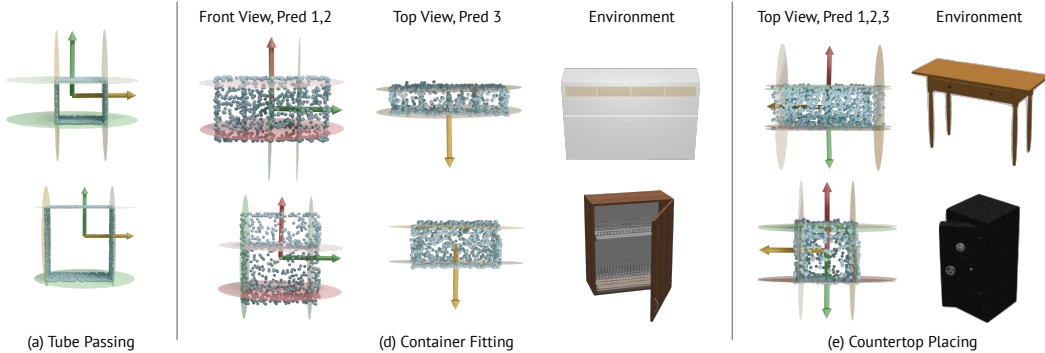

Figure 7: **Geometry Grounding Visualizations.** We plot the learned vectors (as arrows) and planes (as disks) on top of input environment point clouds. We also show the object model next to point clouds for clearer view of object structure. For **(d)** Fit, we visualize predictions in two views for clarity. Please refer to Appendix B.1 for more visualizations.

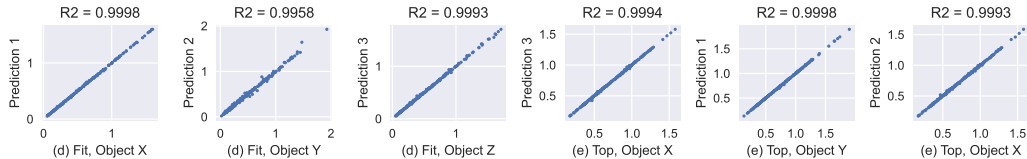

Figure 8: **Improved correlation after using geometry groundings.** We show scatter plots of predicted eigen-length (Y coord.) and their matching "ground truth" (X coord.) in **(d)** Fit and **(e)** Top.

We perform the same correlation analysis and visualize the results in Fig. 8. Compared to Fig. 5, learned eigen-lengths are now almost equal to "groundtruth" thanks to the anchoring effect of the geometry grounding. The extra predicted eigen-length in **(e)** Top also behaves differently, capturing the same "ground truth" as another learned eigen-length. This suggests the regularization from geometry grounding makes learned eigen-lengths more likely to be meaningful measurements. It also reaffirms the fact that the number $S$ of eigen-lengths we set in advance can be different from the actual number of necessary eigen-lengths. Please see Appendix E.4 for detailed discussion.

We also visualize the learned geometry groundings in Fig. 7. The learned vectors align with the main axes of object geometry. The learned planes overlap with tube surfaces in **(a)** Tube, surround the edge of countertops in **(e)** Top, and separate out the region of interest in **(d)** Fit, *e.g.* the higher one out of two storage spaces. These meaningful geometric entities provide a clear interpretation of learned eigen-lengths, *e.g.* in **(e)** Top's case, red and green predictions coincide with each other and both capture the back-to-front length of the countertop.

## 5.3 A STUDY ON THE ROBUSTNESS AND DATA EFFICIENCY OF GEOMETRY-GROUNDED EIGEN-LENGTHS

Geometry grounding of eigen-lengths can be seen as a form of regularization. We are therefore curious how the introduction of geometry groundings may influence the robustness of models in extreme test setups, as well as their data efficiency. We compared the performance of (1) *Direct*, a no-eigen-length approach, where an MLP directly predicts the final label from the concatenation of object and environment latent features. (2) *Implicit*, the minimal eigen-length-based pipeline introduced previously; and (3) *Grounded*, the geometry-grounded version.

Table 1 shows test performances in extreme test conditions with low resolution point clouds or with extraordinary object scalings. Eigen-length-based approaches exhibit much higher robustness.

Fig. 9 shows the trend of test performances as we change the size of training data. We also plot the difference between "ground truth" eigen-length measurement directions (local up and right) and predicted vectors as a way to quantify eigen-length quality. Results suggest that geometry-grounded version is more data efficient if meaningful geometry groundings emerge. When the training data

Table 1: Performance on extreme test cases. All methods are trained on $size = 1024$ point clouds with width $w$ and height $h$ sampled from $U([0.4, 1.0])$.

| | Direct | Implicit | Grounded |
|---|---|---|---|
| Default | 99.00 | 99.15 | 99.65 |
| # Points = 64 | 76.80 | 93.50 | 96.17 |
| $w, h \sim U([0.2, 0.4])$ | 72.10 | 98.78 | 99.72 |
| $w, h \sim U([2.0, 3.0])$ | 82.44 | 98.82 | 99.63 |

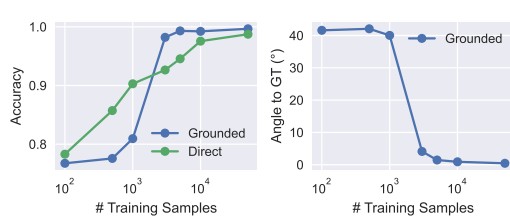

Figure 9: Trend of *Left:* test accuracy and *Right:* average angle between learned vector groundings and "groud truth" directions w.r.t. # training samples.

is limited ($< 3000$ samples), however, the predicted directions of groundings are far from ground truth measurement directions, suggesting that the model fails to learn meaningful groundings for eigen-lengths, and thus the final accuracy is lower than *Direct*.

# 6 CAN EIGEN-LENGTHS BE LEARNED IN MULTI-TASK SETTINGS AND APPLIED TO NEW TASKS?

As humans, we are able to develop a library of useful measurements/eigen-lengths like height from past experience. Given a new task, instead of trying cluelessly, we would start with known measurements and investigate their role in the task. In this section, we ask if learned eigen-lengths can work in a similar way, *i.e.*, given a set of training tasks, is it possible to learn a set of eigen-lengths from them? Further, given a novel task, can we learn to select a subset of learned eigen-lengths that are sufficient for it? In other words, can agents accumulate and transfer knowledge in the form of eigen-lengths?

## 6.1 MULTI-TASK TESTBED

We design a set of tasks that share key eigen-lengths as the testbed for multi-task learning. As shown in Fig.10(a), we consider box-fitting tasks where the box only has a subset of six faces. Each mode of face existence corresponds to a different task with different geometric constraints. For example, to be able to fit, an object has to be narrower than the box in task 2 and shorter than the box in task 3. We set aside the box with all six faces present as the test task. We expect to learn width, height, and length from the training task set, and learn to use all of them during testing. By boxes with partial faces, we aim to mimic different types of cavities in the furniture, e.g., closed drawer as a box with all faces, an open space on the shelf as a box without the front face, etc.

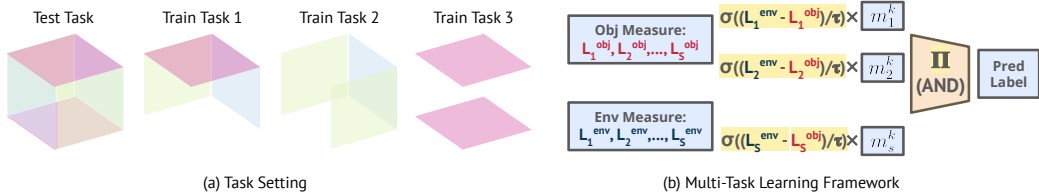

Figure 10: **(a) Multi-Task Setting** where each train task uses boxes with certain faces missing as the environment geometry, and test task uses a complete box; and **(b) Learning Framework**, where we use trainable masks to select eigen-length comparison results.

## 6.2 MULTI-TASK LEARNING FRAMEWORK

Fig. 10(b) shows the multi-task learning framework we experiment with. From a high level, we learn a set of $S$ eigen-lengths and allow each task to select relevant ones from them. This selection step is implemented as a learnable binary mask $\{m_s^k\}_{s=1,2,...,S}$ over eigen-lengths for each task $T_k$. We simply insert the mask in the AND-composition and compute the outcome for $T_k$ as $\prod_{s=1}^{S} m_s^k \cdot \sigma((L_s^{env}(\mathcal{E}) - L_s^{obj}(\mathcal{O}))/\tau)$.

Table 2: Multi-Task learning, novel task adaptation results. We finetune eigen-length-based methods on novel task for 1 epoch and compare them to the direct method trained from scratch for 1 and 100 epochs.

|  | (Single Task) Direct | | (Eigen-Length) Implicit | (Eigen-Length) Grounded |
|---|---|---|---|---|
| Epoch | 1 | 100 | 1 | 1 |
| Test Accuracy | 73.14 | 88.47 | 97.71 | 99.48 |

During training, we optimize both the eigen-length prediction networks and a continuous version of per-task masks $\tilde{m}^k \in [0, 1]$. At test time, we freeze network weights and only learn a mask to choose from eigen-lengths learned during training. Notably, we limit the size of test task data to 10 batches (320 samples) to examine if learned eigen-lengths help in few-shot adaptation scenarios.

### 6.3 MULTI-TASK LEARNING AND FEW-SHOT TEST TASK ADAPTATION

We experiment with both implicit and geometry-grounded eigen-length prediction networks. To analyze the learned eigen-lengths and per-task masks, we visualize learned geometry groundings that are selected ($m_s > 0.5$) in each task in Fig. 11. Meaningful groundings are learned and correctly selected for each task, including the test task.

To explore whether eigen-lengths learned during training help quicker adaptation to new tasks, we compare the test task performance of *Implicit, Grounded* to *Direct* trained from scratch on the test task. All methods are limited to 10 batches of test task samples. As shown in Table 2, within one epoch of finetuning, methods based on the reuse of learned eigen-lengths already achieve high performance, surpassing *Direct* trained from scratch by a large margin, even when the latter has been trained for 100 epochs.

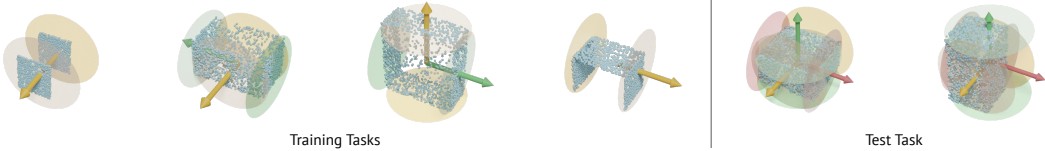

Figure 11: **Learned Geometry Grounding in Multi-Task Setting.** We only show learned geometry grounding (vectors as arrows, planes as disks) selected by the mask in each task.

## 7 CONCLUSION

In this work, we formulate a novel learning problem of automatically discovering low-dimensional geometric eigen-lengths crucial for fitting tasks. We set up a benchmark suite comprising a curated set of fitting tasks and corresponding datasets, as well as metric and tools for analysis and evaluation. We demonstrate the feasibility of learning meaningful eigen-lengths as sufficient geometry summary only from binary task supervision. We show that proper geometry grounding of the eigen-lengths contributes to their accuracy, interpretability, and robustness. We also make an initial attempt at learning shared eigen-lengths in multi-task settings and applying them to novel tasks.

Our exploration suggests broad opportunities in this new research direction and reveals many challenges. For example, grounding eigen-length predictions on geometries requires reasonable choice of geometric primitives, which relies on inductive bias of the specific tasks considered. It would be a challenging future direction to build a universal framework that accommodates a wide range of tasks by leveraging all kinds of geometric primitives and inductive biases. In many task instances, we may have access to signals beyond binary success or failure, *e.g.*, a possible placement position of the object. How to leverage these task signals in eigen-length learning remains an open problem. As a first-step attempt at defining and exploring the challenging problem of eigen-length learning, we do hope our work can inspire more researchers to work on this important yet underexplored direction.

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

## A  IMPLEMENTATION DETAILS

### A.1  NETWORK ARCHITECTURE

The framework in Section 4 consists of a PointNet and an MLP output head that maps the PointNet global feature to $S$ scalar values. The architecture is outlined below, where the numbers in the parenthesis refer to the number of channels in each layer. We use batch normalization and LeakyReLU after all FC layers, except for the output layer.

$$\text{PointNet} \begin{cases} \text{Per-Point MLP}(3 \to 64 \to 128 \to 1024) \\ \downarrow \\ \text{Max Pooling} \end{cases}$$
$$\downarrow$$
$$\text{MLP}(1024 \to 256 \to S)$$

Output: $S$ scalars.

The framework in Section 5 consists of VectorNet and WeightNet. VectorNet consists of a PointNet classification backbone and an MLP output head, as outlined below.

$$\text{PointNet} \begin{cases} \text{Per-Point MLP}(3 \to 64 \to 128 \to 1024) \\ \downarrow \\ \text{Max Pooling} \end{cases}$$
$$\downarrow$$
$$\text{MLP}(1024 \to 256 \to 3S)$$

Output: $S$ vectors.

WeightNet consists of a PointNet segmentation backbone and $2S$ parallel MLP output heads, each outputs a weight distribution over all points, as outlined below.

$$\text{PointNet} \begin{cases} \text{Per-Point MLP}(3 \to 64[\text{per-point feature}] \to 128 \to 1024) \\ \downarrow \\ \text{Max Pooling}[\text{global feature}] \end{cases}$$
$$\text{Concat(per-point feature, global feature)}$$
$$\downarrow$$
$$\text{MLP}((1024 + 64) \to 512 \to 256 \to 128)$$
$$\downarrow$$
$$\text{Output Weight MLP}_i(128 \to 256 \to 1), i = 1, 2, \ldots, 2S$$
$$\downarrow$$
$$\text{SoftMax}$$

Output: $2S$ sets of per-point weights.

We use LeakyReLU and batch normalization after each FC layer except for the output layers.

### A.2  TRAINING DETAILS

All networks are implemented using PyTorch and optimized by the Adam optimizer, with a learning rate starting at $10^{-3}$ and decay by half every 10 epochs. Each batch contains 32 data points; each epoch contains around 1600 batches. We train models for $\sim 100$ epochs on all tasks. The learnable parameter $\tau$ is initialized with $\tau = 1$. All experiments are run on a single NVIDIA TITAN X GPU.

### A.3  DATASET DETAILS

Table 3 and 4 summarizes the statistics of environment/object shapes used in our dataset. Each shape is drawn with probability in inverse proportion to the number of shapes in its category, such that each object category appears with similar frequency in the final dataset.

Table 3: Environment Shape Statistics.

|       | Box | Microwave | Refrigerator | Safe | Storage Furniture | Table | Washing Machine | Total |
|-------|-----|-----------|--------------|------|-------------------|-------|-----------------|-------|
| Train | 21  | 9         | 34           | 21   | 272               | 70    | 13              | 440   |
| Test  | 7   | 3         | 9            | 7    | 73                | 25    | 3               | 127   |

Table 4: Object Shape Statistics.

| Train | | | | | | | | |
|--------|--------|------|-----|-----|-----|-----|----------|-------|
| Basket | Bottle | Bowl | Box | Can | Pot | Mug | TrashCan | Total |
| 77     | 16     | 128  | 17  | 65  | 16  | 134 | 25       | 478   |

| Test | | | | |
|--------|-----------|-----|--------|-------|
| Bucket | Dispenser | Jar | Kettle | Total |
| 33     | 9         | 528 | 26     | 554   |

During data generation for the tasks where both the environment and the object are ShapeNet objects, we apply random scaling $s \sim U([0.9, 1.1])$ to the environment objects, set all joints to closed state and sample $M = 1024$ points from the object model. Given an object-environment pair, we randomly sample $T = 1000$ candidate positions in the environment point cloud, and check whether placement of the object at each candidate satisfy the task specification using SAPIEN (Xiang et al., 2020) simulation. If all candidates fail, we label the pair as negative, otherwise as positive. Specifically, the candidate positions are sampled from "applicable and possible regions" following Mo et al. (2021b)'s definition. For example, we only consider points with upward facing normals, and for task (e) only consider points with close to highest z coordinates. We generated around 75K training data and 20K testing data for each task.

## B  ADDITIONAL RESULTS

### B.1  GEOMETRIC GROUNDING VISUALIZATION AND FAILURE CASE DISCUSSION

Fig. 12 and 13 show more visualizations of the learned eigen-lengths in the three tasks from the main paper. Our framework is able to learn reasonable eigen-lengths that measure along crucial directions. These eigen-lengths are also grounded by planes that suggest the relevant part of object which supports the task. In experiments with primitive shapes as environments, the learned planes almost overlap with the box/tube faces. In experiments with ShapeNet container objects as environments, especially task (d) (*Fit*, or container fitting) as shown in Fig. 13, locating the relevant part becomes more challenging. As this usually involves finding cavities in a shape and selecting the largest one. Fig. 13 shows examples of our learned eigen-lengths, most of which make sense, as shown in (a)-(o). We are able to ignore irrelevant parts, e.g. the legs of tables, and find the part of object that affords the "containment" task, e.g. the drawer in (b), the closet in (c). When there are many cavities that afford the same task, the network picks the largest one, e.g. in (d) and (k).

**Failure Cases.**   We also observe some failure cases where the learned eigen-lengths are inaccurate. Fig. 13(p)-(t) shows the most representative ones. (p) shows a relatively complex shape, where the network struggles to find the correct width of the drawer. (q) and (r) show cases where the network finds the wrong cavity. According to our task definition, the object can only be placed in the drawer part in (q). Instead, the network finds the part on top of the drawer. In (r), the network finds the second largest cavity instead of the largest one at the bottom. (s) shows an extreme case where the height of the pizza box is much smaller than the other two extents. As objects usually have correlated extents, comparing height suffices most of the time. The network probably lacks the motivation to precisely capture the width and the length of the pizza box, resulting in the underestimation of width and length in (s). Finally, our formulation, i.e. the AND clause of three eigen-length comparisons, can not fully and precisely describe the nature of this task. The washing machine in (t) has a cylinder-shaped cavity, which our network tries to approximate by a cuboid, which is reasonable within the range of its expressive power but not accurate. Also, there could be shapes that do not have a "largest"

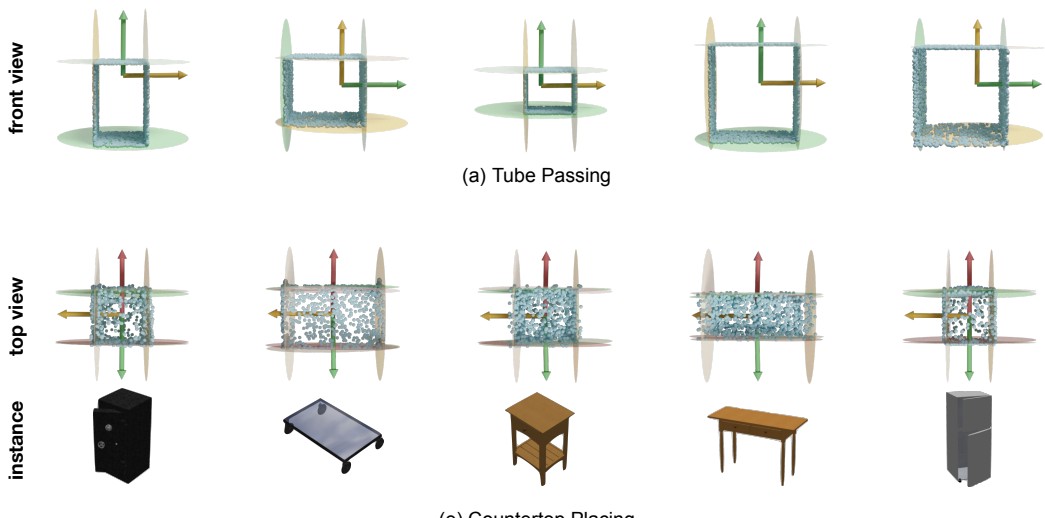

Figure 12: **Additional qualitative results.** We visualize the learned vectors and planes for (a) Tube Passing and (e) Countertop Placing. We show all eigen-lengths in the front(a)/top(e) view. We also show the underlying instances in task (e) countertop placing for a clearer understanding of the object structure. Note that though some joints are "open" for visualization purpose, all instances in the dataset are at their rest state.

cavity, e.g. some drawers in a closet may be designed for tall and narrow things, while others are designed for flat things. To deal with arbitrary objects, the extents of both types of drawers are useful. Introducing more complex and flexible formulations, e.g. in Section D, would help better capture the complexity of the task.

### B.2 CORRELATION ANALYSIS RESULTS

We show here the scatter plots and correlation $R^2$ values between all prediction eigen-lengths and all presumable geometric measurements. $R^2$ value, or coefficient of determination, is a metric in $[0, 1]$ reflecting linear correlation between two variables. The closer $R^2$ is to 1, the more linearly correlated the two variables are. Given two set of samples $x_i, y_i$, where $i = 1, 2, \ldots, n$, $R^2$ is defined between $y_i$ and the least squares linear regression of $y_i$ on $x_i$, $\tilde{y}(x_i)$:

$$R^2 = 1 - \frac{\sum_i (y_i - \tilde{y}(x_i))^2}{\sum_i (y_i - \bar{y})^2},$$

where $\bar{y} = \frac{1}{n} \sum_i y_i$ is the mean value of $y_i$.

Results from *Eigen-Length-Implicit* are shown in Fig. 14. Results from *Eigen-Length-Grounded* are shown in Fig. 15. We can clearly see the one-to-one correspondence between predictions and presumable measurements. $R^2$ is close to or greater than $0.9$ where the prediction is the match for the measurements, otherwise the value is much smaller. It is more apparent in the *Eigen-Length-Grounded* variant, where $R^2$ values are close to the theoretical bound 1 when it matches. The models can learn a compact and appropriate set of eigen-lengths from binary task supervision. Also note that the extraneous prediction slot in task (e) (*Top*, or countertop placing) become degenerate with another prediction slot, as has mentioned before in main text.

## C APPLYING RANDOM ROTATIONS TO INPUT ENVIRONMENT GEOMETRIES

### C.1 FORMULATION AND IMPLEMENTATION

While the object shape is randomly rotated in all experiments in the main paper, we take environment geometry directly from ShapeNetChang et al. (2015) where shapes are axis-aligned. In this section, we consider a more challenging setting where the environment geometry is also randomly rotated.

Specifically, we consider a "rotated" version of the Container Fitting task and Tube Passing task in the main paper. For each original data point, i.e. a container/tube-object pair with a boolean label ((object point cloud $P_o$, environment point cloud $P_e$), success label $L$), we sample a random rotation $R$ and apply it to both the container and the object. We feed $(RP_o, RP_e)$ to the network described in Section 5 and supervise the network with the same label $L$.

## C.2 CORRELATION ANALYSIS AND RESULT VISUALIZATION

We show the correlation analysis of learned eigen-lengths in Fig. 16. A strong, disentangled correlation between learned eigen-lengths and human-hypothesized ones can still be observed.

We also visualize the learned geometry groundings in Fig. 17. The predicted planes roughly align with the main cavities of the objects. From the results, we can see that the proposed problem setting is still valid and the studied methods can still be applicable and produce reasonable results.

# D EXTENDING AND CLAUSES TO DISJUNCTIVE NORMAL FORM (DNF)

## D.1 FORMULATION

We employ the AND clause formulation for all tasks shown in the main paper. Namely, after learning a library of paired object/environment eigen-lengths $\{(L_s^{env}, L_s^{obj})\}_s$, we compose them by

$$\hat{T}(\mathcal{E}, \mathcal{O}) = \bigwedge_{s=1,2,...,S} [L_s^{env}(\mathcal{E}) > L_s^{obj}(\mathcal{O})],$$

(selection mask $m$ is omitted for clarity), approximated by

$$\tilde{T}(\mathcal{E}, \mathcal{O}) = \prod_{s=1,2,...,S} \sigma((L_s^{env}(\mathcal{E}) - L_s^{obj}(\mathcal{O}))/\tau).$$

Here we show we can extend this formulation to the more general Disjunctive Normal Form (DNF), where an OR connects multiple AND clauses. Each AND clause composes eigen-length comparison results of a subset of eigen-lengths. The result of each AND clause is then aggregated by an OR operator. More precisely,

$$\hat{T}(\mathcal{E}, \mathcal{O}) = \bigvee_{U_a \in \mathcal{U}} \bigwedge_{s \in U_a} [L_s^{env}(\mathcal{E}) > L_s^{obj}(\mathcal{O})].$$

$\mathcal{U} = \{U_a\}_a$ specifies the subset $U_a$ of eigen-lengths in each AND clause. We similarly use a differentiable approximation during training:

$$\tilde{T}(\mathcal{E}, \mathcal{O}) = 1 - \prod_{U_a \in \mathcal{U}} (1 - \prod_{s \in U_a} \sigma((L_s^{env}(\mathcal{E}) - L_s^{obj}(\mathcal{O}))/\tau)).$$

The introduction of two-level logic and the OR operator helps express more complex reasoning and deal with a wider range of tasks. For example, many realistic tasks have multiple solutions. OR captures the relationship that the task can be executed if any, not necessarily all, of the solutions work.

## D.2 TASK AND IMPLEMENTATION DETAILS

To demonstrate our framework's compatibility with this new formulation, we experiment with the *Multi-Tube Passing* task. This is a variant of task (a) (*Tube*, or tube passing) in the main paper, where we have two tubes of random sizes placed next to each other. As long as the object can be translated and passed through any of these tubes, the task is considered as successful.

Similar to tube passing, we randomly sample the extents of the tubes, the shape, scale, and rotation of the object. The center of the two tubes are always at two fixed positions on the $y$-axis.

We set the number of eigen-lengths to learn as $S = 4$ and split them into two disjoint AND groups, namely $\mathcal{U} = \{\{1, 2\}, \{3, 4\}\}$. Ideally, the learned eigen-lengths should correspond to the height and width of the tubes. Also, the height and width of the same tube should be in the same AND group.

### D.3 RESULT VISUALIZATION

Fig. 18 visualizes the learned eigen-lengths, where green and yellow belong to one group, purple and red belong to another group. We successfully learn eigen-lengths that measure along the height/width directions of the tubes. We also learn them in correct groups, where width and height of the same tube are paired together.

## E DISCUSSION AND FUTURE WORK

### E.1 DEFINITION OF EIGEN-LENGTHS AND APPLICATION SCOPE OF THE EXPLORED FRAMEWORK

In our setting, an eigen-length is whatever scalar measurement (i.e., just a 1D scalar) the network invents to best perform its stated downstream task. While this definition for eigen-dimensions is quite general and could be applicable to any object as long as there exist certain 1D eigen-lengths that are crucial and useful for checking the feasibility of accomplishing a downstream task, we are assuming in our current experiments that having such sets of 1D eigen-lengths are *sufficient* for the tasks. Therefore, our current setting would not apply to the tasks where having only such low-dimensional eigen-lengths is not sufficient, such as the tasks of geometric contour matching and object collision checking.

### E.2 BROADER IMPLICATION OF THE STUDIED APPROACH FOR AI AND ROBOTICS

We believe the general approach we suggest can have very general applicability in AI and robotics, where the solution to downstream tasks suggests the emergence of generally useful geometric concepts such as length, height, width, and radius in unsupervised ways. As we described in the introduction, learning such compact useful geometric eigen-lengths is beneficial in the ways that 1) they are highly interpretable, while most of the current learned representations in neural networks are opaque and learned as black-box hidden features which may be unreliable or untrustworthy, 2) they could be shared and reused across different tasks, enabling fast adaptation to novel test-time tasks, and 3) the proposed learning formulation may discover novel yet crucial geometric eigen-lengths that are even unknown to us humans given the new test-time tasks. Furthermore, there could be more geometric concepts of great interest and importance that future work can explore in this direction. Examples can be 1) symmetry, as a result of trying to complete 3D shapes, 2) regular object arrangements and poses as a tool for efficient search, and 3) tracking, as an essential capability for predicting the outcome of sports games. In other words, we want learning networks to invent the notions so symmetry, regularity, or tracking. If such capabilities could emerge from purely unsupervised learning, we no longer need to rely on black-box-like neural networks and human annotations for this geometric information over 3D objects.

### E.3 ROTATION OF OBJECTS DURING TASK EXECUTION

In our experiments, we are primarily concerned with translational motion during task execution. This setting stems from practical concerns: in many robotic manipulation scenarios, the rotation of the object is often given as the desired target to achieve by robot planners or unchangeable during robotic grasping and manipulation. For example, the robot gripper may only be able to grasp, hold and move the mug without spilling the content and with steady grasping in certain poses for a pick-and-place task and the robot may not be able to freely rotate the object as the arm kinematics may not allow.

That being said, our cylinder fitting task does allow rotation along the up-axis for the object and similarly, the sphere fitting task allows the full $SO(3)$-space rotation, while for other tasks, in the case that multiple poses of the object are possible, we can simply pass the object in different poses into the same network for multiple times to query the joint fitting feasibility.

### E.4 DETERMINING THE NUMBER OF EIGEN-LENGTHS TO LEARN

The number of eigen-lengths to learn, i.e. $S$, is a hyperparameter of our learning framework and has to be set in advance. However, it should be interpreted as the upper bound on the number of

eigen-lengths the system can learn, and does not have to be the "groundtruth" number of relevant eigen-lengths. As shown in Sec. 4.3 and Sec. 5.2, when we set $S = 3$ for the countertop fitting task where only two eigen-lengths matter, the extra "slot" either degenerates or coincides with other slots. Such cases can be easily detected and filtered, and the actual number of relevant eigen-lengths can be discovered. Setting a maximum number for an unknown number of targets is also a common practice in problems like object detection Redmon & Farhadi (2017). That being said, a more flexible mechanism that allows an arbitrary number of eigen-lengths would be desirable, especially for objects with complex compositional structures like robotic arms or closets with many drawers. We leave this as a future direction.

## F  NEGATIVE SOCIAL IMPACT

Our work joins the initial efforts of eigen-length emergence in unsupervised learning settings with many other works along this direction. There could be several bias in the data and training objectives, but this is a general concern shared by most works in this field. Our work may also share the controversial arguments with other works that the future AI agents may have the capability of thinking by themselves causing threats for human beings. However, the results demonstrated in our work are way far from that. Other than these potential issues mentioned above, we do not see any other major concerns our work particularly introduces.

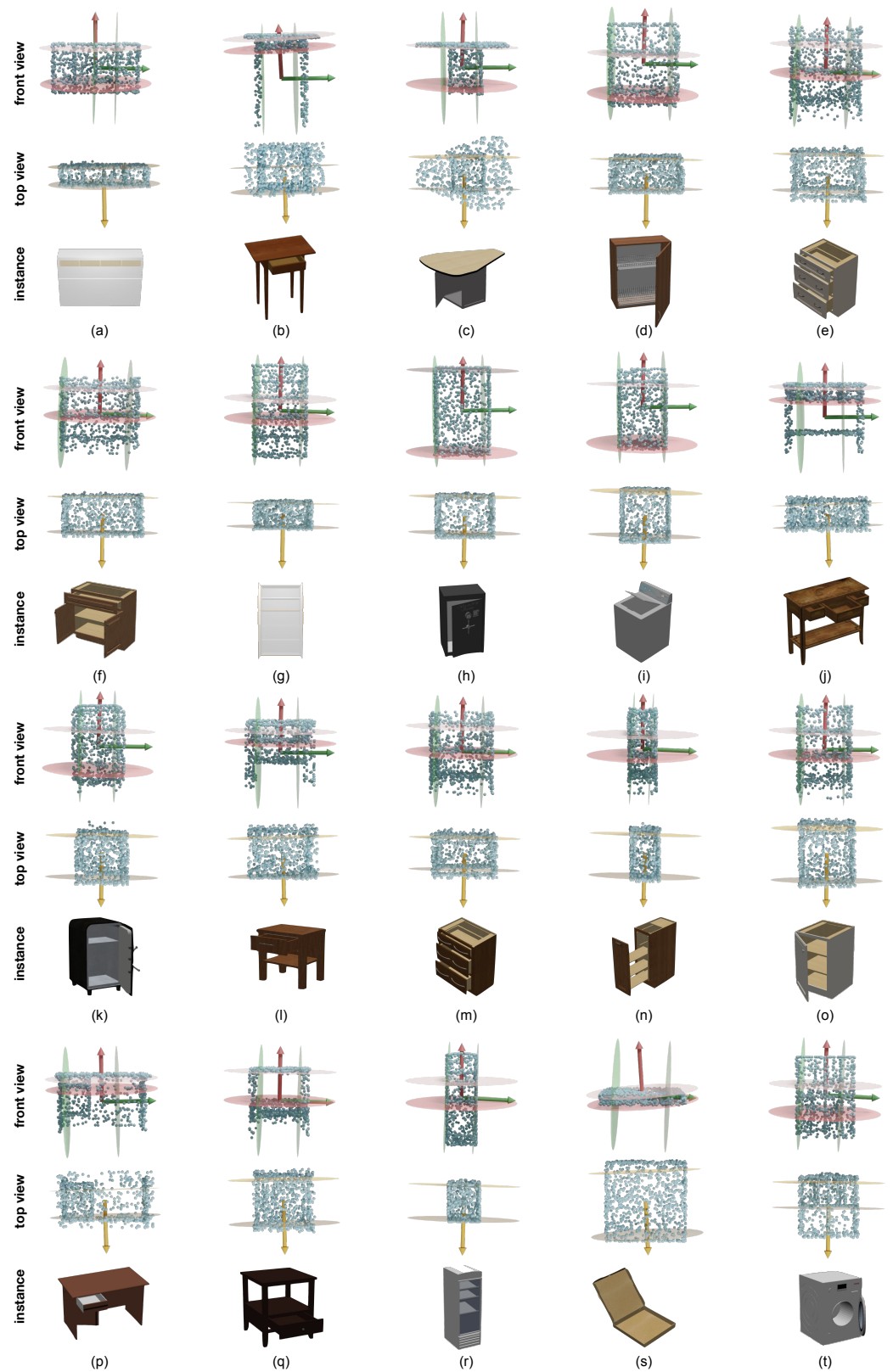

Figure 13: **Additional qualitative results in Container Fitting**. We show eigen-lengths in two views together with the underlying object following Fig. 12 (d). (a)-(o) are successful cases where the learned planes correctly separate out the largest cavity in the object. (p)-(t) show failure cases.

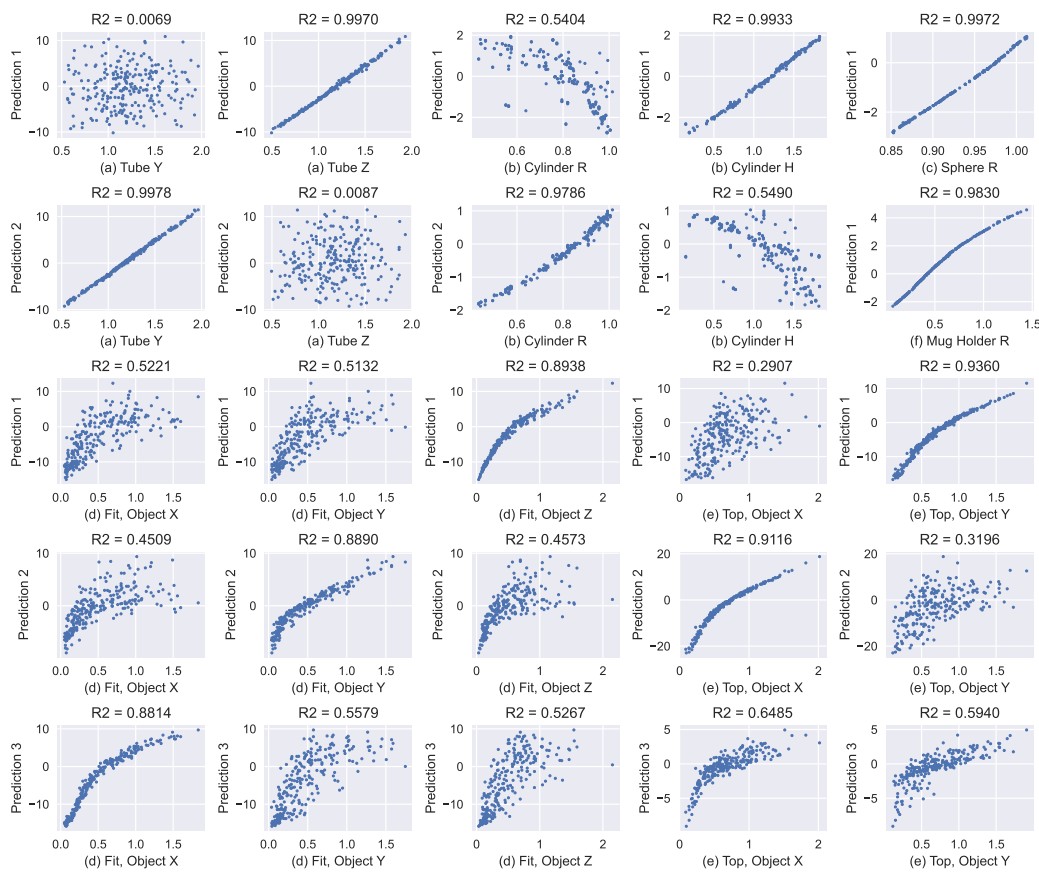

Figure 14: **Full correlation plots and respective $R^2$ values** between ground truth measurements and predicted eigen-lengths from *Eigen-Length-Implicit*.

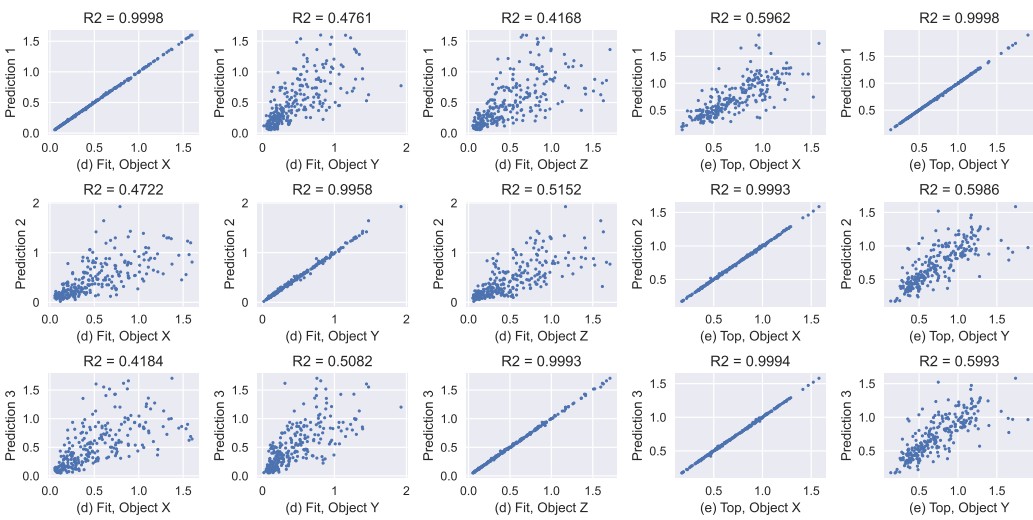

Figure 15: **Full correlation plots and respective $R^2$ values** between ground truth measurements and predicted eigen-lengths from *Eigen-Length-Grounded*.

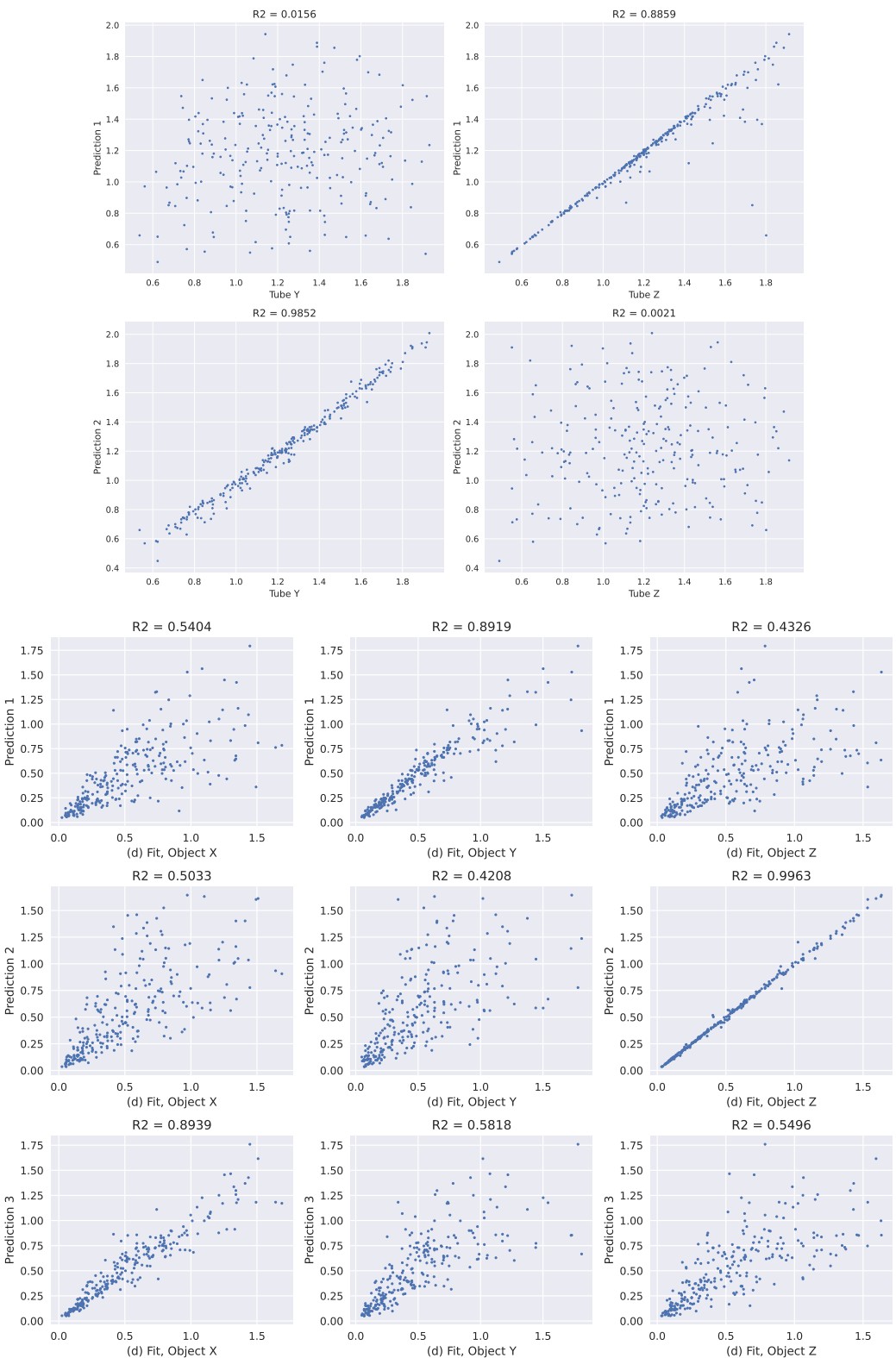

Figure 16: **Full correlation plots and respective** $R^2$ **values** between human-hypothesized measurements and predicted eigen-lengths in rotated Tube Passing and rotated Container Fitting, respectively. Correspondences between predicted eigen-lengths and human-hypothesized ones can be observed.

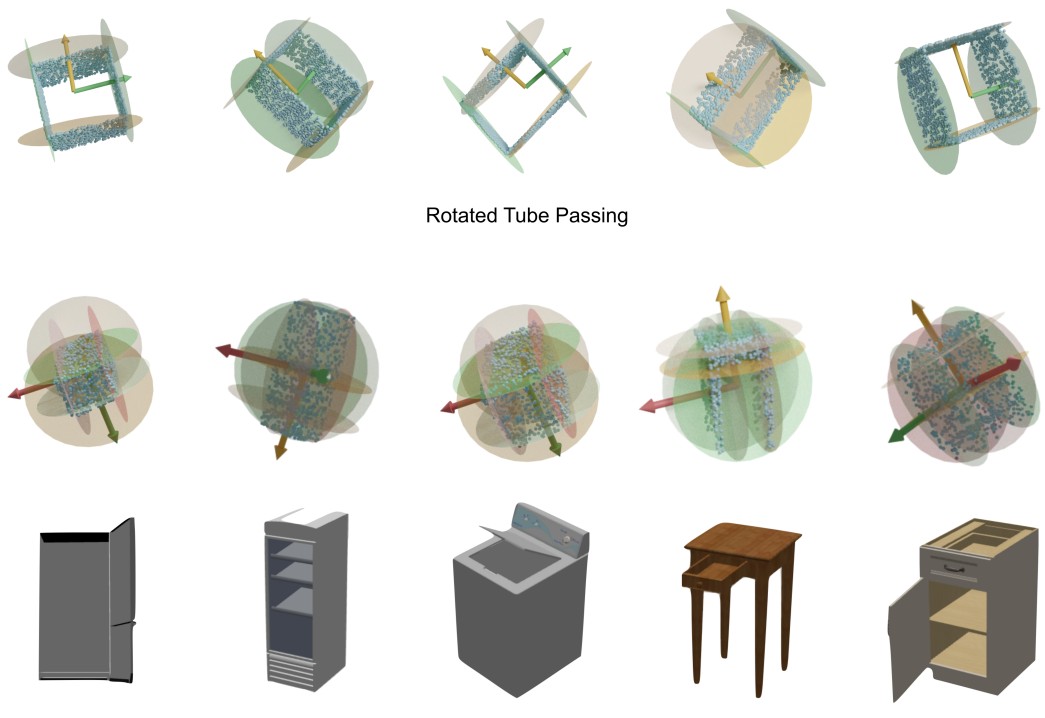

Figure 17: **Visualization of learned geometry groundings** in rotated Tube Passing and rotated Container Fitting, respectively. Vectors are visualized as arrows, and planes are visualized as disks. For Container Fitting, we also show the underlying geometry (before rotation) for better reference. The learned vectors and planes roughly align with the rotated object. Regions of interest like drawers are also selected by planes.

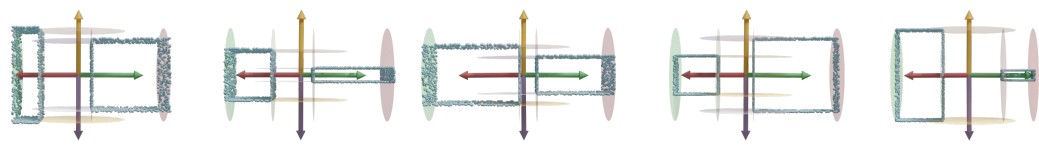

Figure 18: **Visualization of the eigen-lengths learned with OR-AND clauses**. Green and yellow, purple and red eigen-lengths belong to the same AND-group. It turns out that each group attends to one of the tubes and captures its width and height.

