# OpenReview forum: "Toward Learning Geometric Eigen-Lengths Crucial for Robotic Fitting Tasks"
_ICLR.cc/2023/Conference — Submitted to ICLR 2023_

### Official Review · Reviewer_rLUp · 2022-10-22

**Confidence:** 3
**Correctness:** 3
**Technical Novelty And Significance:** 2
**Empirical Novelty And Significance:** 2
**Recommendation:** 3

**Clarity, Quality, Novelty And Reproducibility:**

###### Clarity
- The article is well-written, and the authors express their thoughts very clearly.
- However, for some statements, the paper does not provide enough results to verify them, such as lacking the comparison between different kinds of representations.
- The texts in figures could be larger for better readability.

###### Quality
- The quality is good. I believe the authors spent a lot of time and effort to build this benchmark.

###### Novelty
- The novelty is limited.

###### Reproducibility
- If the authors release the benchmark, the results will be easy to reproduce.


**Strength And Weaknesses:**

###### Pros
- The idea of learning task-specific geometry lengths of the object and the environment is interesting.
- This paper is written clearly, with the motivation and methodology easy to follow.
- The authors build a comprehensive benchmark suite including 6 kinds of tasks, ~1200 object models from ShapeNet, the corresponding datasets, and evaluation metrics.
- The findings of being able to learn length directly from binary supervision are interesting and impressive.

###### Cons
- To apply the eigen-length learning step to robot fitting or other robot manipulation problems, it seems that how to choose and define the geometric primitive and the creteria is essential to this problem. After that step, parameter learning is relatively easy. Therefore, whether the proposed problem setting and method are practical remains unclear. This method design is contrary to the goal of the paper, which aims to rely on the least human prior.
- To predict geometric information, there are tons of ways, including traditional image processing methods like edge detection, deep learning methods over computer vision like 3D object detection[1], segmentation[2], 6 DoF pose estimation[3] and computer stereo vision system[4] with necessary pre-process and post-process operations. The proposed method for the robot fitting task is one of the extended applications of the geometric information prediction problem, and thus the novelty is limited.
- There are no experimental comparisons to baselines. For example, other robot fitting methods (or simpler, binary-classification methods) that explicitly estimate or implicitly learn the geometric information should be compared; Comparisons of the low-dimensional eigen-length to the high-dimensional latent codes and structured representations discussed in the Introduction section should also be empirically verified.
- The constraints of the problem setting, i.e., only translation (without rotation) of pre-aligned objects and environments is too strict to be feasible in practice.
- For Table 2, should the proposed method be compared with other transfer learning methods instead of direct training without knowing the training tasks?


###### Refs
- [1] Zhou, Yin, and Oncel Tuzel. "Voxelnet: End-to-end learning for point cloud based 3d object detection." Proceedings of the IEEE conference on computer vision and pattern recognition. 2018.
- [2] Wu, Bichen, et al. "Squeezeseg: Convolutional neural nets with recurrent crf for real-time road-object segmentation from 3d lidar point cloud." 2018 IEEE International Conference on Robotics and Automation (ICRA). IEEE, 2018.
- [3] Xiang, Yu, et al. "Posecnn: A convolutional neural network for 6d object pose estimation in cluttered scenes." arXiv preprint arXiv:1711.00199 (2017).
- [4] Wikipedia: Computer stereo vision. https://en.wikipedia.org/wiki/Computer_stereo_vision

**Summary Of The Paper:**

This paper introduces a new problem setting that learns the geometric eigen-lengths for robot fitting tasks. The authors build a benchmark for this task, propose two learning methods (learning directly from binary supervision and learning with geometric primitive), and test the methods with single-task and multi-task(transfer learning) settings.

**Summary Of The Review:**

This paper proposes an interesting topic, and it is clear to follow.
However, the validity of the problem setting and the comparisons to relevant methods should be addressed more carefully.

---

### Official Review · Reviewer_VXuJ · 2022-10-25

**Confidence:** 4
**Correctness:** 4
**Technical Novelty And Significance:** 3
**Empirical Novelty And Significance:** 3
**Recommendation:** 8

**Clarity, Quality, Novelty And Reproducibility:**

This is a very interesting problem, tackling the interesting question of how to what mode of description of 3D objects mediates common-sense reasoning. I found the paper was well written, clear and interesting to read. The proposed approach is, to my knowledge, novel.

The description of the experiments is clear and well detailed, but it would be important for reproducibillity that the code and data be made available.


**Strength And Weaknesses:**

### Strengths
- The paper is well written and pleasant to read.
- This is an interesting problem. The extraction of intrinsic properties of objects such as eigen-dimensions is an interesting and important avenue of research.
- The use of the multi-task learning and task adaptation is well adapted to the problem.
### Weakness
- Although the general concept of eigen-dimensions is intuitive and several examples are discussed, the article lacks a formal definition of eigen-dimensions and their meaning for a range of objects would be useful to clarify the applicability of the approach.
- The broader implication of the results for AI and robotics should be made explicit. What is the possible usage outside of the narrow tasks described in this paper?


**Summary Of The Paper:**

This article is concerned with the notion of eigen-lengths of objects, canonically width, height and depth. The authors argue that those eigen-lengths are key to common-sense reasoning (eg, can object A fit in container B). The authors propose that eigen-lengths should be grounded in the scene geometry.
The paper attempts to ground the eigen-lenths using a numbe of manipulation tasks.
Experiments demonstrate that the approach allowed for accurate estimation of the eigen-lengths.

**Summary Of The Review:**

In summary, I found this paper  interesting and pleasant to read. The approach is novel and tackling an important problem for the community. I did not find any major concerns on this paper, although some of the discussion could be improved to clarify the contribution (see weaknesses above)

---

### Official Review · Reviewer_rqrT · 2022-10-25

**Confidence:** 3
**Correctness:** 2
**Technical Novelty And Significance:** 3
**Empirical Novelty And Significance:** 2
**Recommendation:** 5

**Clarity, Quality, Novelty And Reproducibility:**

The concepts are presented clearly, but sometimes at a level that is too abstract.

Failure cases and limitations are discussed in the supplement.

The work is original, to the best of my knowledge.
Reproducing the methods based on the descriptions in the main text and supplement is not an easy task. The use of PointNets for classification and segmentation is briefly mentioned without many other details.

Minor Comment: No publication venues are provided for both papers by Higgins et al.


**Strength And Weaknesses:**

Assessing the strengths and weaknesses of this paper is difficult because the problem addressed is under-explored in the literature and the solutions presented here are not mature yet. This seems to be an interesting direction for research, but there are several shortcuts and limitations in the formulation. While this is partially understandable, I have doubts whether this is the right direction and whether extensions of this formulation are likely to be adopted by other researchers.

Strengths

This is an interesting problem and a thought-provoking paper presenting a novel take on it.

The claim that the eigen-lengths are interpretable and universal is true, but also self-evident. Determining if an object fits in different objects depends on it dimensions. I do not consider this a major strength.

The concept of grounding, as defined in the paper, leads to improved performance.

The comparison of the direct, implicit and grounded methods is informative, but see below about my concerns on the representation.

Weaknesses

The restriction of object motion to translation only appears reasonable for a first attempt at the proposed formulation, but reduces the search for “eigen” lengths to essentially the selection of relevant dimensions of the bounding box. Since both objects and containers/tubes are axis-aligned, the problem is greatly simplified and does not require the learning of affordances other than the concept of support surfaces in some cases. (It is not clear to me if points of contact are detected in the sphere fitting task, for example.)

Success or failure of a given task is determined based on the existence of suitable final placement of the object in the environment. In some cases of interest, the existence of a feasible trajectory is also needed. An object may fit in a cabinet or drawer but there may be no way to put it in there.

The fact that the number of relevant eigen-lengths, S, has to be set manually is a limitation.

The chosen representation of the objects and environments requires further justification. PointNet descriptors are not well suited for capturing object properties such as dimensions. It seems that some other representation, possibly in the form of random projections, would be more effective.

Section 6.1 is hard to connect to a practical scenario. (It is possible that I am missing something.)

The test set in Table 4 is dominated by jars. I do not see any reason to construct an imbalanced dataset for an unexplored problem.


**Summary Of The Paper:**

The paper explores an under-investigated problem, that of automatically discovering geometric quantities that indicate the potential success or failure of fitting tasks. These include passing an object through a tube-like container, putting an object in a drawer or a sphere, etc. The authors use the term eigen-lengths for these measurements and explore three approaches for learning them from data, with the help of some manual annotation in the form of specifying the number of relevant eigen-lengths. Several tasks accompanied with relevant data are defined and quantitative results are presented. (Presumably, the data and test suite will be released if the paper is accepted.)

**Summary Of The Review:**

On one hand, accepting the paper may inspire other researchers to make progress on an interesting problem, surpassing the baseline algorithms presented here. On the other hand, some of the limitations listed above are serious. I recommend borderline accept for now to give the benefit of the doubt to the authors, but justification for the chosen representations, the use of PointNet, and the restriction to translation-only motion should be provided in the rebuttal. Section 6 should be explained more clearly.

++++++++++++ Dec. 11 update ++++++++++++++++++

After the discussion among the reviewers and area chair and the latest author responses, I am leaning towards recommending rejection. Even though the idea is interesting, I had undervalued the limitations, which outweigh the contributions. The handling of rotation in particular is unsatisfactory. According to the authors’ latest response, rotations in rectangular environments cannot be handled. This is an important limitation. I also do not understand the value of rotating spheres or sphere-shaped environments when the representation is purely geometric. Moreover, the response on the ordering of eigen-lengths so that they correspond across the object and environment is unsatisfactory. I do not see how the network can learn that based on the loss function.

---

### Official Review · Reviewer_4TGL · 2022-10-26

**Confidence:** 3
**Correctness:** 4
**Technical Novelty And Significance:** 2
**Empirical Novelty And Significance:** 2
**Recommendation:** 5

**Clarity, Quality, Novelty And Reproducibility:**

There are no major issues related to the clarity of this paper as the idea is simple and intuitive.

**Strength And Weaknesses:**

Strengths
----------

Ideas are clearly elaborated through the paper with adequate support of intuition in explanations.

Elaboration of experimentations is adequate for an early exploration of learning object lengths.

Initial results are promising on the capability of learning lengths from trials.


Weaknesses
-------------
This paper is an early exploration of the topic. Consequently, the experimentation is limited to a single dataset and simplistic tasks. Can this methodology be used to improve challenging tasks such as relative pose estimation and active visual navigation?


**Summary Of The Paper:**

This paper presents an early attempt at learning geometric lengths from trails and explores how to transfer the learned geometric knowledge to solve a different task. Besides offering a problem formulation, this paper presents an evaluation framework based on finding evaluation correlations against ground truth lengths.

**Summary Of The Review:**

Overall this is initial research. My main criticism is the lack of exploration or at least discussion on how to use this idea in more complex problems. I would like to see at least some discussion in this regard.

---

### Comment · Area_Chair_TyNj · 2022-12-06
**Request for Clarification**

Dear Authors,

During the discussion with the reviewers, a few questions came up that we were hoping to get clarification on. Of particular note:

1. In Section 4.2 ObjNet and EnvNet output S-dimensional vectors $\overrightarrow{L^\textrm{obj}}$ and $\overrightarrow{L^\textrm{env}}$ for the objects and environments, respectively. Task success is measured according to whether for each index s, the entry in the environment vector is greater than that in the object vector, i.e., $\overrightarrow{L_s^\textrm{env}} > \overrightarrow{L_s^\textrm{obj}}$. In this case, the ordering of elements in each vector is important as it specifies the alignment between $\overrightarrow{L_s^\textrm{obj}}$ and $\overrightarrow{L_s^\textrm{env}}$. **How is this ordering determined?**
2. The last paragraph on the "Fitting with Rotations" page of project website (https://sites.google.com/view/geometric-eigen-length/fitting-with-rotations?authuser=0) referenced in the author response states that:

 > Given an arbitrarily large number $N$, we can similarly construct more complex shapes with more rectangles such that the shape cannot be described by $2N$ numbers. To sum up, when fitting arbitrary shapes into rectangles allowing rotations, there isn't a bounded set of eigen-lengths sufficient to describe all shapes, therefore, our setting does not apply."

The author response to the reviewers' comments/questions regarding the (in)ability to handle rotations states that the method is able to handle full rotations about the up-axis for the cylinder and sphere fitting tasks. **Can you clarify whether it is correct to conclude from the paragraph above that there are shapes for which the method can not handle rotation and provide further insight as to what differentiates these objects from those for which the method can deal with rotations?**

3. In order for the results to be reproducible and for the proposed benchmark to be usable by other researchers, the reviewers agree that the dataset and code would need to be made public. Unless we are missing it, we do not see a reference to making the dataset and code public in the paper or in the author response. **Is it your intention to make the dataset and code publicly available?**

Best,
AC

---

### Decision · Program_Chairs · 2023-01-20

**Decision:**

Reject

**Justification For Why Not Higher Score:**

The problem setting itself is of little value and the proposed solutions are quite limited. In the AC's opinion, the limitations outweigh the contributions, and the AC believes that most of the other reviewers would agree.

**Justification For Why Not Lower Score:**

N/A

**Metareview: Summary, Strengths And Weaknesses:**

The paper considers the problem of learning low-dimensional geometric embeddings for robot fitting tasks. The paper proposes a new benchmark for robot fitting and investigates different strategies for learning these features, which the paper refer to as "eigen-lengths". Using the proposed benchmark, these strategies are evaluated for different single- and multi-task learning settings.

The paper received four reviews that are largely in agreement that the problem of learning task-relevant representations for robot fitting tasks is interesting. However there was notable disagreement among the reviewers regarding the significance of the paper's contributions with regards to the benchmark and proposed solutions. In particular, Reviewer VXuJ finds the approach to be novel, while two of the other reviewers (rqrT and rLUp) emphasize the limitations of the formulation in light of what the reviewers find to be a overly constrained problem setting and strong assumptions, notably regarding the (in)ability to handle rotations and the differences between the learned representation and traditional geometric features (e.g., bounding boxes). Because of this discrepancy in the initial overall recommendations, Reviewers VXuJ, rqrT, and rLUp met (virtually) with the AC to discuss the merits of the paper. As a result of the discussion, questions were posed to the authors in an attempt to get clarification about the method's generalizability to rotations, as well as the reproducibility of the work. The authors' response suggests that the practicality of the solutions are quite limited due to the inability to handle rotations when the shapes are rectangular. It was then questioned why rotations are important in the context of spherical objects. Future versions of the paper would benefit from a clear discussion of the differences between the learned representations and vanilla geometric features as well as the extent to which the methods generalize to different object shapes.

Note: The review provided by 4TGL is short on details. The reviewer did not reply to or acknowledge the authors' response, nor did they participate in the virtual reviewer-AC discussion. As a result, less weight was placed on their review.

**Summary Of Ac-Reviewer Meeting:**

Reviewers VXuJ, rqrT, and rLUp met (virtually) with the AC for an hour to discuss the merits of the paper. The following are notes from the meeting.

* The word "release" doesn't appear in the paper and the authors didn't respond to the reviewers' question
* Also thinks that the method is novel
* rLUp: Problem is definitely under-explored. Paper only focuses on robot fitting (reviewer feels that it is a narrow form of robot fitting, yet paper over-claims)
* rLUp: Doesn't agree with authors' claim that their method is unsupervised. It requires knowledge (i.e., a label) of whether the object can be fitted (the binary label, which the reviewer feels is hard to get)
    ** rqrT: This could be acquired in simulation and may well generalize to the real world
    ** VXuJ: The difference is that the supervision that the paper assumes is different from the baselines (which would require bbox labels)
    ** rLUp: Seems that the formulation does not extend to general object geometries (mentioned on the website)
    ** rqrT and VXuJ: Don't buy the argument that rotation is not possible for many fitting tasks, but then again, authors show that rotation is possible with the cylinder
* rqrT/VXuJ/rLUp: The paper definitely has flaws, but it may inspire future work. It is not a 1% improvement on what is known
* rqrT: When the eigen-lengths are inferred, how does the approach know the alignment between the object and environment lengths (relevant for the constraint that L_s^env > L_s^obj --> how do the authors know that L_1^env matches L_1^obj if they just get an unordered set of lengths)
    ** rqrT: This section (4.2) is a core part of the paper and would have benefited from more details. Not sure whether this is a deal-breaker
    ** rqrT: In the case that the eigen-lengths are bounding box dimensions, this ordering might come out of PointNet, but its not clear what they do when the lengths are not bbox (which the authors emphasize may be the case.)
    ** rLUp: The results are impressive, but without knowing how to train the model, the results may not be reproducible or the method usable for other domains.
    ** VXuJ: This is why the code should have been released
** rqrT: Not convinced that the eigen-lengths in the sphere (symmetric) domain do not correspond to bboxes.
** VXuJ (and AC): It's not obvious why the eigen-length doesn't correspond to the bbox for the cylinder either. It is clear for the mug, though.
** rqrT:  Website says that "a bounded set of eigen-lengths sufficient for the task does not exist" yet method requires that the size of this set be bounded (the value S)
* rqrT: The method does not model whether or not a trajectory exists that fits the object in the container
* rqrT: Website states that "To sum up, when fitting arbitrary shapes into rectangles allowing rotations, there isn't a bounded set of eigen-lengths sufficient to describe all shapes, therefore, our setting does not apply." suggesting that the method does not allow for rotation.
* rqrT/rLUp: Difference from methods that learn task-relevant embeddings is that this embedding is very low-dimensional and geometric in nature.
* rqrT: Was borderline positive, but could go borderline negative depending on their response.
* VXuJ: Was positive, but is less supportive if they don't release the dataset/code

Ask the authors for clarification regarding:
    ** The last sentence on the Website regarding the ability to handle rotations
    ** The ordering/alignment (section 4.2)
    ** Whether or not they plan to release the data/code